# *Phlebotomus papatasi* Antimicrobial Peptides in Larvae and Females and a Gut-Specific Defensin Upregulated by *Leishmania major* Infection

**DOI:** 10.3390/microorganisms9112307

**Published:** 2021-11-06

**Authors:** Barbora Kykalová, Lucie Tichá, Petr Volf, Erich Loza Telleria

**Affiliations:** Department of Parasitology, Faculty of Science, Charles University, Viničná 7, 12800 Prague, Czech Republic; barbora.kykalova@natur.cuni.cz (B.K.); luc.ticha@email.cz (L.T.); volf@cesnet.cz (P.V.)

**Keywords:** sand fly, insect immunity, gut-specific response, defensin, *Leishmania*

## Abstract

*Phlebotomus papatasi* is the vector of *Leishmania major*, causing cutaneous leishmaniasis in the Old World. We investigated whether *P. papatasi* immunity genes were expressed toward *L. major*, commensal gut microbes, or a combination of both. We focused on sand fly transcription factors dorsal and relish and antimicrobial peptides (AMPs) attacin and defensin and assessed their relative gene expression by qPCR. Sand fly larvae were fed food with different bacterial loads. Relish and AMPs gene expressions were higher in L3 and early L4 larval instars, while bacteria 16S rRNA increased in late L4 larval instar, all fed rich-microbe food compared to the control group fed autoclaved food. Sand fly females were treated with an antibiotic cocktail to deplete gut bacteria and were experimentally infected by Leishmania. Compared to non-infected females, dorsal and defensin were upregulated at early and late infection stages, respectively. An earlier increase of defensin was observed in infected females when bacteria recolonized the gut after the removal of antibiotics. Interestingly, this defensin gene expression occurred specifically in midguts but not in other tissues of females and larvae. A gut-specific defensin gene upregulated by *L. major* infection, in combination with gut-bacteria, is a promising molecular target for parasite control strategies.

## 1. Introduction

Phlebotomine sand flies (Diptera: Psychodidae) belonging to *Lutzomyia* and *Phlebotomus* genera are proven vectors of *Leishmania* parasites (reviewed by [1]), causing 700000 to one million new cases of leishmaniasis every year [2]. *Phlebotomus papatasi* is dispersed across Mediterranean European countries, North Africa, the Middle East, and Central Asia. It transmits *Leishmania major*, one of the etiological agents of cutaneous leishmaniasis (reviewed by [3]). Nevertheless, migration and environmental changes constantly shape the ecoepidemiology of leishmaniases [2].

During the cycle in the sand fly vector (reviewed by [4,5]), *Leishmania* parasites coexist with a diverse microbial community that may interfere with the parasite establishment in *P. papatasi* [6,7,8,9,10,11,12,13]. Concomitantly, the sand fly immune response is adjusted to the presence of commensal and other possible harmful microbes (reviewed by [14]).

In insects, the activation of the Toll and immune deficiency (IMD) pathways occurs when transmembrane receptors, such as Toll-like receptors and peptidoglycan recognition proteins (PGRP), recognize pathogen-associated molecular patterns (PAMPs) [15]. Once these receptors are activated, a sequence of intracellular signaling events occurs, involving regulatory proteases and kinases, resulting in the translocation of transcription factors (e.g., dorsal and relish) to the nucleus and transcription of effector molecules such as antimicrobial peptides (AMPs) [15].

Regarding sand flies, the IMD pathway is involved in the response to *Leishmania*. For instance, the upregulation of this pathway in *Lutzomyia longipalpis*, through the knockdown of its repressor caspar, reduced *Leishmania infantum* and *Leishmania mexicana* survival [16]. In addition, the depletion of the pathway through the knockout of the transcription factor relish resulted in an increase in *L. major* and gut bacteria in the *P. papatasi* gut [17]. The sand fly AMPs are potentially responsible for a deleterious effect on the parasite. For example, a recombinant defensin peptide encoded by the *Phlebotomus duboscqi* defensin gene has effective activity against *L. major* promastigotes [18]. Moreover, the suppression of a defensin gene in *L. longipalpis*, mediated by RNAi, slightly increased *L. infantum* detection [19]. These results suggest that sand flies express defensins potentially driven toward the parasites.

These studies showed that the IMD pathway and AMPs could affect both *Leishmania* and gut bacteria. Nevertheless, it is not yet clear whether the parasite *per se* triggers the sand fly immune response and if such a response would be triggered specifically in the sand fly gut. In the present study, we focused on analyzing the effects of microbe-rich larvae food with a particular interest in the expression of genes mediated by Toll and IMD pathways. In addition, we depleted the gut bacterial community of adult females to assess the *P. papatasi* female immune response to *L. major*. Our results provide information on *P. papatasi* expression of dorsal and relish transcription factors and attacin and defensin AMPs.

## 2. Materials and Methods

### 2.1. P. papatasi Immunity Genes

Two transcription factor sequences that contain the rel homology domain (RHD) were selected [20]. The *Phlebotomus papatasi* relish transcription factor gene was previously identified [17]. Dorsal transcription factor, attacin, and defensin were identified by similarity using *L. longipalpis* sequences [21] as a query to search on the *P. papatasi* RNAseq database publicly available from the Vector Base website (www.vectorbase.org, accessed on 10 September 2018), using blast search tools. Partial coding sequences were amplified by PCR using *P. papatasi* cDNA as a template and were sequenced. Similarities between *P. papatasi* sequences and other insect vectors were assessed by the MUSCLE multiple sequence alignment tool [22] built-in Geneious 7.1.9 software (Biomatters, Auckland, New Zealand). This was followed by phylogram analysis using the Maximum Likelihood method with a bootstrap value of 400 repetitions in MEGA X 10.0.5 software [23]. The best substitution model was estimated using MEGA X software using the lowest Bayesian Information Criterion (BIC) score. The Whelan and Goldman (WAG) model [24] was used in dorsal, relish, and defensin analyses, while the Le_Gascuel (LG) model [25] was used in the attacin analysis.

### 2.2. L. major Culture

*Leishmania major* parasites (FV1 MHOH/IL/80/Friedlin) were cultivated in Medium 199 (Sigma–Aldrich, Saint Louis, MI, USA) at 23 °C, supplemented with 10% heat-inactivated fetal bovine serum (Thermo Fisher Scientific, Carlsbad, CA, USA), 1 % BME vitamins (Sigma–Aldrich), 2 % of sterile urine, and 250 μg/mL amikacin (Medopharm, Pozorice, Czech Republic). Propagation of *L. major* promastigote culture had up to five passages before the sand fly experimental infections.

### 2.3. P. papatasi Colony Rearing

The *Phlebotomus papatasi* colony, established from sand flies caught in Turkey in 2005, was maintained under standard conditions [26]. Larvae were kept in plastic pots filled with plaster of Paris and fed larvae food made from composted rabbit feces. Three- to seven-day-old females were fed anesthetized mice and were transferred to plaster-lined pots for oviposition four or five days after blood-feeding. Larvae and adult insects were kept at 26 °C.

### 2.4. Larvae Experimental Feeding

Larval food made from composted rabbit feces [26,27] was divided into two parts, one part was sterilized in an autoclave, and another part was kept unaltered, henceforth referred to as microbe-rich food. Both were collected from the same batch of composted food; therefore, they had same initial composition. We did not make an identification of the bacteriome present in the types of larvae food used in our experiments. Both autoclaved (control group) and microbe-rich food were kept at 4 °C until use. Rearing pots from the two different feeding regimens were kept separately, and fresh food was added three times a week. Observation of larvae development and emerged adults was recorded following colony maintenance routine three times per week. A suspension of each type of larvae food was plated on Luria Bertani (LB) agar medium and incubated for 48 h at 25 °C to estimate the bacterial load. Larvae of the second (L2), third (L3), and early- and late-fourth (L4) instars were dissected, and the guts were collected in pools of 15 individuals from each instar. The experiment was repeated three times. First instar larvae were not sampled due to their diminutive size.

### 2.5. Depletion of Sand Fly Gut Bacteria

An antibiotic cocktail (AtbC) composed of 100 units/mL of penicillin (BB Pharma, Martin, Slovakia), 50 μg/mL of gentamicin (Sandoz, Boucherville, Canada), and 4 μg/mL of clindamycin (Sigma–Aldrich)*,* adapted from [28], was used to deplete the bacteria community of sand fly female guts. The AtbC was added to 30 % sucrose solution, and 100 μL of the mixture was offered to the recently emerged females *ad libitum* in small Petri dishes. The AtbC-sucrose mixture was changed daily during the experiments.

Bacterial depletion was checked after one week of AtbC treatment on sucrose meal (prior blood-feeding) and five days post blood-feeding of the control and experimental groups. Insects were surface-cleaned twice in a 70 % ethanol bath and rinsed in a sterile saline solution before dissections. Pools of 10 dissected guts from AtbC-treated or non-treated females were homogenized in 100 μL of fresh sterile saline solution and plated on a blood-agar medium. Colony-forming units (CFUs) were counted after 48 h of incubation at 25 °C.

### 2.6. Leishmania Experimental Infection

One control (blood-fed, non-infected) and two (infected) experimental feeding groups were prepared using seven-day AtbC-treated *P. papatasi*. The control group was fed blood and sucrose meal, both containing AtbC. Female sand flies in the experimental groups were fed defibrinated sheep blood (LabMediaServis, Jaromer, Czech Republic) seeded with 10^6^ *L. major* promastigotes/mL. In one experimental group, AtbC was added to the infected blood meal and to the sucrose meal offered *ad libitum* to the sand flies after infectious feeding. A second experimental group received the infected blood meal and sucrose meal without AtbC. The suppression of AtbC allows bacteria to recolonize the sand fly gut (Table 1).

Sand fly samples were collected at different time points post blood-feeding. Guts and corresponding carcasses were stored at −70 °C in lysis buffer until RNA extraction.

### 2.7. RNA Extraction, cDNA Synthesis, and PCR

Total RNA was extracted from pools of 10 dissected guts or carcasses using the High Pure RNA Tissue Kit (Roche, Pleasanton, CA, USA) according to the manufacturer’s instructions. The RNase-free DNase I (Thermo Fisher) digestion step at 1 U/μg of RNA was used to clear possible genomic DNA traces. Synthesis of cDNA was carried using a Transcriptor First Strand cDNA Synthesis Kit (Roche) following the manufacturer’s instructions.

Conventional PCR targeting the *P. papatasi* actin gene (Table 2) was used to test successful cDNA synthesis. Reactions were done according to the EmeraldAmp GT PCR Master Mix (TaKaRa, Shiga, Japan) instructions. Cycling conditions were as follows: 95 °C for 3 min; 34 amplification cycles (95 °C for 30 s; 60 °C for 30 s, 72 °C for 1 min); and 72 °C for 5 min. The same conditions were used with defensin primers (Table 2) for detection in different tissues. Amplicons were visualized on 1.5 % agarose gel.

### 2.8. Relative Gene Expression by qPCR

Expression of sand fly immunity genes, *Leishmania* actin, and bacteria 16S rRNA were determined by qPCR in a LightCycler 480 thermocycler (Roche) with gene-specific oligonucleotides (Table 2) and SYBR Green I Master (Roche). The cycling conditions were as follows: 95 °C for 10 min enzyme activation, 45 amplification cycles (95 °C for 10 s, 60 °C for 20 s; 72 °C for 45 s). Relative gene expression was calculated in comparison to the *P. papatasi* reference gene actin (PPAI004850-RA) and ribosomal protein L8 (PPAI008202-RA) and expressed as the fold change in comparison to the autoclaved-fed or blood-fed control groups [29].

### 2.9. Leishmania Infection Estimation and Morphometrics

*Leishmania* development in sand fly vectors was examined by light microscopy in 20 sand fly guts 144 h post infection (day 6 PI), a time when the blood meal was digested and defecated. Under the conditions used, most *P. papatasi* females defecated on day 4 PI [30]. Guts were dissected in saline solution (NaCl 0.9 %), covered with a thin glass slide, and examined under a 40× magnification objective lens. Parasite abundance was estimated and classified as low (less than 100 parasites), moderate (between 100 and 1000 parasites), or heavily infected (more than 1000 parasites) [31]. The localization of parasites in the gut (abdominal or thoracic gut, cardia, and colonized stomodeal valve) was recorded to evaluate infection progress, following previously published methods [32]. Samples were collected from two independent experiments.

Parasite developmental stages were assessed by morphometric methods. Images of 250 randomly selected promastigotes were captured from sand fly gut smears on Giemsa-stained glass slides under a 100× magnification objective lens. Cell width, length, and flagellum were measured using the microscope scale plugin in ImageJ 1.52a software [33]. *Leishmania* developmental stages were categorized as procyclic promastigotes (body length < 14 μm and flagellar length ≤ body length), elongated nectomonads (body length ≥ 14 μm), metacyclic promastigotes (body length < 14 μm and flagellar length ≥ 2× body length), and leptomonads (short nectomonads = remaining parasites) [32,34]. Samples were collected from two independent experiments.

**Table 2 microorganisms-09-02307-t002:** Oligonucleotides.

Reference	Gene	Sequence
Nadkarni et al. 2002 [35]	Bacteria 16S rRNA	5′ TCCTACGGGAGGCAGCAGT 3′
		5′ GGACTACCAGGGTATCTAATCCTGTT 3′
Di-Blasi et al. 2015 [36]	*Leishmania* actin	5′ GTCGTCGATAAAGCCGAAGGTGGTT 3′
		5′ TTGGGCCAGACTCGTCGTACTCGCT 3′
(PPAI004850)	*P. papatasi* actin	5′ GCACATCCCTGGAGAAATCCTAT 3′
		5′ GGAAAGATGGCTGGAAGAGAGAT 3′
(PPAI003791)	*P. papatasi* attacin	5′ GCCATTTCTGCTGCGTACTC 3′
		5′ GAGGCACCAAGTACACGACA 3′
(PPAI004256)	*P. papatasi* defensin	5′ GCCCGGTTAAAGACGATGTAAAG 3′
		5′ AGTTGGTCCAAGGATATCGCAAG 3′
(PPAI001149)	*P. papatasi* dorsal	5′ GCTGCAAATCCTGCAAAGA 3′
		5′ CCCAAGGAGGTCACAGGTTA 3′
Louradour et al. 2019 [17]	*P. papatasi* relish	5′ ATCCATCCTTTATGCAACCG 3′
		5′ GCCTTTGAGTCGCAGTATCC 3′
(PPAI008202)	*P. papatasi* ribosomal protein L8	5′ GACATGGATACCTCAAGGGAGTC 3′
		5′ TTGCGGATCTTATAGCGATAGGG 3′
VectorBase gene identification shown in parenthesis.		

### 2.10. Statistical Analysis

Ordinary two-way ANOVA with Sidak’s correction for multiple comparisons test built-in GraphPad Prism software (version 6.07) (GraphPad Software Inc., San Diego, CA, USA) was used to calculate significant differences in gene expression results obtained by qPCR, and infection estimation and localization were obtained by light microscopy observation.

## 3. Results

### 3.1. Transcription Factors and Antimicrobial Peptide Genes

The two *P. papatasi* transcription factors belong to the nuclear factor-kappa B (NF-κB) superfamily, namely, dorsal (PPAI001149) and relish (PPAI012820). The dorsal amino acid sequence contains a rel homology domain (RHD) and a rel homology dimerization domain (RHDD) (Figure 1A). The relish amino acid sequence also contains an ankyrin repeat domain in addition to RHD, RHDD (Figure 1B). The *Phlebotomus papatasi* dorsal sequence shares close similarity to the *L. longipalpis* dorsal sequence forming a sister branch with the *Aedes* and *Anopheles* dorsal sequences (Figure 1C). The relish sequence was closely grouped with the *L. longipalpis* sequence and formed a sister clade with sequences from *Aedes* and *Culex* identified as NF-κB (Figure 1C).

The two *P. papatasi* AMPs identified by similarity with the *L. longipalpis* genes are attacin (PPAI003791) and defensin (PPAI004256). The attacin deduced amino acid sequence contains the corresponding superfamily domain (Figure 2A). The defensin sequence contains the Defensin-2 superfamily domain (Figure 2B). The phylogenetic analysis showed that *P. papatasi* defensin (PPAI004256) grouped with *L. longipalpis* defensin 4 (LlDef4), but it was separated from a clade containing *L. longipalpis* defensin 2 (LlDef2) and *P. duboscqi* sequences (Figure 2C).

For all *P. papatasi* genes used in our experiments, we designed gene-specific primers within each coding sequence used in PCR amplifications. Amplicons were sequenced to confirm the targeted gene. In our current approach, we were not able to distinguish differences between *P. papatasi* sequences derived from VectorBase (sand flies originated from Israel) and from our colony (originated from Turkey).

### 3.2. Expression of Immunity Genes in Larval Guts

We wanted to determine whether the selected *P. papatasi* immunity genes would be expressed throughout development and altered under different diet conditions in guts dissected from various larval instars. The larval growth period was slightly delayed in the group fed autoclaved food between the L3 and early-L4 stages, but no difference was observed between the early-L4 and late-L4 stages (Appendix A). Nevertheless, there was no noticeable difference in the larvae size and survival rates, neither in the size or number of emerged adults between the two groups of reared larvae. When we plated a suspension of food samples on LB agar, we observed no bacterial growth in the sample of autoclaved food while massive/significant bacterial growth was present in the microbe-rich food sample.

There was no significant modulation in the gene expression of dorsal in the group fed microbe-rich food compared to the group fed autoclaved food (Figure 3A). On the other hand, the expression of other immunity molecules was increased in larvae fed microbe-rich food. Particularly, relish was increased in the early L4 stage (Figure 3B), attacin was increased in the L3 and early L4 stages (Figure 3C), while defensin was increased in the L3 stage (Figure 3D). The relative gene expression of bacterial 16S increased significantly in the late L4 stage (Figure 3E).

### 3.3. Expression of Immunity Genes in Infected Females with Depleted Gut Bacteria

We first tested the efficiency of AtbC in blood-fed sand flies. We evaluated the CFUs from dissected guts from sand flies five days post blood-feeding, when all females had eliminated the digested blood content. The CFUs were significantly reduced in AtbC-treated in comparison to the non-treated group (Appendix A).

We also tested if AtbC treatment would interfere with the parasite development in the sand flies. On the sixth day post infection, we assessed the intensity and progression of infection in AtbC-treated (+AtbC) sand flies compared to the non-treated (-AtbC) group. There was no significant difference in infection intensity and localization levels in the sand fly gut (Figure 4A,B). In addition, we analyzed the parasite developmental forms on gut smears, and we did not observe significant differences between the +AtbC- and -AtbC groups (Figure 4C).

We hypothesized that the sand fly immune response could be specifically induced by *L. major*. To address this possibility, we used *P. papatasi* sand flies and assessed the expression of dorsal, relish, attacin, and defensin genes in females infected by *Leishmania* (experimental group 1–EG1) compared to the non-infected control group. Both control and experimental groups were AtbC-treated before and after experimental feeding.

In dissected guts of the *Leishmania*-infected group EG1, dorsal expression was increased at 48 h while relish showed no significant expression changes compared to the AtbC-treated blood-fed control group (Figure 5A,B). Attacin did not show significant changes while defensin expression significantly increased at 144 h post infection (Figure 5C,D). *Leishmania* detection showed that it increased at 72 h post infection compared to the parasite loads detected on the first day post infection (Figure 5E).

### 3.4. Immunity Genes and Infection Progression in Infected Females with Recovered Gut Bacteria

To test if the recolonization of gut bacteria would modify the immune response in the *Leishmania* infected sand fly gut, we removed the AtbC treatment from a group of sand flies (experimental group 2–EG2).

We assessed the effect of removing AtbC on bacterial detection in the sand fly guts. We seeded gut homogenates on blood agar plates and observed an increase in CFUs at 72 h and 144 h post infection in the recovered bacterial group compared to the AtbC-treated group (Appendix A). This increase was also detected by the relative expression of bacterial 16S rRNA (Appendix A).

When we compared the EG2-infected group with recovered gut bacteria to the blood-fed control group, there were no significant differences in the expression levels of the transcriptional factors dorsal and relish (Figure 6A,B). At 72 h post infection, attacin expression was variable, therefore showing a non-significant increase (Figure 6C); nevertheless, defensin was significantly increased at the same time point (Figure 6D). The expression of the *Leishmania* actin gene used to assess the parasite detection showed that the parasite increase at 72 h was quite variable, with no significant difference compared to the parasite loads 24 h post infection (Figure 6E).

We also compared the gene expression data of *Leishmania*-infected sand flies with recovered bacteria (EG2) compared to the infected group constantly treated with AtbC (EG1). The relative gene expressions of all five analyzed genes were not significantly altered (Appendix A).

### 3.5. Defensin Gut-Specific Expression

We investigated if the *P. papatasi* defensin gene was also expressed in other sand fly tissues in larvae and adult females. qPCR detected defensin amplification only in guts and not in other tissues. Therefore, we performed non-quantitative PCR using dissected tissues, followed by electrophoresis in 1.5 % agarose gel for visual representation. In larvae, we observed defensin amplification in the guts of various instars but not in carcasses (Figure 7A). In blood-fed females, defensin expression was found in midguts dissected 24 and 144 h post blood meal (Figure 7A). However, no defensin expression was observed in other tissues such as the head, thorax, Malpighian tubules, ovaries, or posterior end of the abdomen of blood-fed females (Figure 7B).

## 4. Discussion

We selected two transcription factors genes belonging to Toll and IMD pathways (dorsal and relish) and two AMPs (attacin and defensin) to tackle the questions we raised regarding the *P. papatasi* immune response toward the changes in gut bacteria and *L. major*.

We searched for the gene sequences in public databases and used their predicted amino acid sequences to identify signature domains to support gene identification. *Phlebotomus papatasi* dorsal and relish have RHD and RHDD domains characteristic of the NF-κB superfamily [37,38]. The relish sequence contains a C-terminal ankyrin-repeat domain that is characteristic of the ‘NF-κB protein’ sub-family. Dorsal belong to the ‘rel protein’ sub-family sequences that lack the ankyrin-repeat domain [39,40,41]. Both *P. papatasi* dorsal and relish sequences are similar to those previously identified in *L. longipalpis* [21]. They form a group closely related to mosquitoes in both cases while less similar to other flies such as *Drosophila* and *Stomoxys*. Phylogenetic analyses of these transcription factors commonly show mosquito sequences forming a separate branch from *Drosophila* species [42] and more distantly related to other insects such as *Nasonia vitripennis* and *Tribolium castaneum* species [43]. These findings suggest that these sand fly and mosquito transcription factors evolved from a common ancestor.

The *P. papatasi* AMPs attacin and defensin sequences have the signature domains of their respective protein families. The attacin family signature domain contains a proline-rich propeptide (N-terminus) and two glycine-rich domains (C-terminus) [44]. In a previous study, the *P. papatasi* attacin sequence was shown to be closely related to *L. longipalpis* and *Nyssomyia neivai* [19]. In addition, the dipteran attacin group formed separately from the coleopteran and lepidopteran groups [45]. The insect defensins motif has six conserved cysteines responsible for intra-chain disulfide bonds [46]. It was also previously reported that the sand fly defensin group was divided into two branches, one containing the *L. longipalpis* defensin 1, 3, and 4, (LlDef1, LlDef3, and LlDef4) and another containing the LlDef2 and *N. neivai* defensin 2 [19]. Interestingly, the *P. duboscqi* defensin was previously shown to be closely related to the black soldier fly *Hermetia illucens* [47], and in our current analysis, it grouped with the *L. longipalpis* defensin 2. On the other hand, the *P. papatasi* defensin sequence investigated in the present study (PPAI004256) is closely related to the LlDef4, forming a separate branch from the *P. duboscqi* defensin. Together, these studies show that sand fly defensins form diverse groups, but it is unclear which selection pressures acted on their diversification.

In insects, feeding on an enriched microbe food triggers the expression of a complex gene set. For instance, in the larvae of *Trichoplusia ni* moth, feeding on a food mixture containing *Micrococcus luteus* (Gram-positive) and *Escherichia coli* (Gram-negative) can trigger the expression of a group of effector molecules involved in the larvae immune response including AMPs [48]. Similarly, in the larvae of *L. longipalpis*, feeding on *Bacillus subtilis* (Gram-positive) or *Pantoea agglomerans* (Gram-negative) also modulates genes coding for receptors, regulators, and effector molecules of immunity pathways [49]. Therefore, we explored the effect of ingested microbes present in the larvae food on the *P. papatasi* immune response. We cannot exclude possible effects on larvae immunity derived from nutrient processing or absorption caused by the differences in the microbial composition of the food [50]. Nevertheless, our experimental approach is supported by the fact that both food types originated from the same batch, thus having the same initial nutrient composition. In addition, there was no noticeable difference in development between the two larval groups.

Dorsal, the transcription factor of the Toll pathway, showed no increased expression under the microbe-rich larvae rearing while relish, associated with the IMD pathway, was upregulated in the actively eating L4 stage. Under this experimental condition, we cannot rule out the participation of the Toll pathway in the regulation of *P. papatasi* AMPs, but its contribution may be reduced. On the other hand, the IMD pathway has a more prominent role through relish expression.

Attacin was highly expressed in the *P. papatasi* L3 and early L4 stages fed microbe-rich food. These stages are voracious [49,51], thus indicating that attacin is necessary for controlling the increased ingestion of bacterial content in the food. Curiously, defensin was increased in the L3 stage but not in early L4. This may have occurred as a counterbalance between effector molecules or due to changes in bacterial diversity within the larvae gut. Similarly, *L. longipalpis* larvae adjust their AMPs expression throughout their developmental stages with a more evident increase in attacin and two defensin genes (LlDef2 and LlDef4) in L3 larvae compared to the non-feeding larval stage [19]. In addition, AMPs expression in *L. longipalpis* L3 larvae was also adjusted according to the microbial challenge offered through artificial feeding. The ingestion of *B. subtilis* or *P. agglomerans* reduced attacin as early as 12 h, while *P. agglomerans* increased LlDef1 at 24 h post feeding [49]. Together, these findings indicate that sand fly larvae adjust the AMPs expression to balance various bacteria. Sand fly attacins may be responsible for balancing loads of general ingested bacteria, while defensins are tuned to compose a more refined response. This balance is crucial for sand flies since their breeding sites are rich in microorganisms and decomposing material [52,53,54].

Our study was not focused on changes in bacterial diversity. Nevertheless, we observed that the overall bacterial load was slightly increased in *P. papatasi* L3 and early L4, indicating that the sand fly immune response, possibly through attacin expression, controlled the bacteria abundance in these larval stages. In the late L4 stage (a non-feeding stage), the detection of bacteria was highly increased in the group fed microbe-rich compared to the group fed autoclaved food, thus suggesting that a given bacterial population resists after the peaks of AMPs. These remaining bacteria may survive transstadially and contribute to the colonization of the gut in the adult stage, similar to *P. duboscqi* that carried *Ochrobactrum* sp. from the larvae to the pupae and adult stages [55]. Therefore, the efficient balance of the microbial community in the larvae gut can interfere directly in the adult stage, with a possible impact on sand fly fitness.

We treated the adult sand flies with AtbC before the parasite infection to deplete their gut bacteria. In previous studies, AtbC alone showed no deleterious effect on the parasites *Leishmania donovani* and *L. infantum* or the sand fly *L. longipalpis* survival [28,56]. In our initial trials, AtbC did not cause negative effects on *L. major* culture or *P. papatasi* survival. Nevertheless, changes in the vectors’ gut microbial environment may have a distinct outcome. The reduction of bacteria interfered negatively with the progression of *L. infantum* infection in *L. longipalpis*, evidenced by the reduction of metacyclic promastigotes [28]. Under our conditions, our choice of AtbC did not eliminate bacteria but significantly reduced them. In addition, the progression of *L. major* infection in *P. papatasi* evaluated at the sixth day post infection was not significantly altered in the AtbC-treated group. In both groups, parasites multiplied and migrated anteriorly from the abdominal midgut to the thoracic part and the stomodeal valve, and a small percentage of metacyclic forms could be detected in both AtbC-treated and non-treated groups, as observed in previous studies [32,57]. Indeed, the induced variation of the gut microbiota may offer a considerable challenge to the parasite, but the outcome of this balance reveals the potentials of the *Leishmania* adaptability.

In the context of sand fly and *Leishmania* interactions, it is relevant to highlight that the ingested parasites remain inside the *P. papatasi* peritrophic matrix (PM) during the early phase of infection [58,59]. The PM poses a physical barrier between the parasite and its vector, and it could level down the effect of the parasite over the sand fly immune response. Nevertheless, we did not rule out the possibility that secreted parasite molecules and exosomes [60,61,62] could affect the sand fly immune response during the early phase of infection when the PM is formed and then degraded. Therefore, we analyzed the *P. papatasi* gene expression before and after PM degradation.

We used the bacteria-depleted sand fly model to investigate the sand fly immune response to *L. major*. One experimental group and the control group were AtbC-treated throughout the experiment. The experimental group was infected by *L. major*, while the control group was blood-fed. Dorsal was upregulated in infected sand flies at a time when attacin was highly variable, and this points to a possible connection between this transcription factor and the effector molecule. Relish showed no significant modulation except for a slight increase on the sixth day when defensin was upregulated, which may also indicate the connection between these molecules. *Phlebotomus papatasi* attacin was slightly but not significantly downregulated at three out of four time points analyzed, indicating that the *L. major* infection caused this subtle reduction. Nevertheless, this variability may also be a result of an under-detected variation of AtbC-resistant bacteria. Moreover, attacin was possibly regulated by another pathway such as Jak-STAT, as similarly reported in *Drosophila* [63,64].

In our experiments, the most significant difference in gene expression levels happened with defensin. The upregulation of this AMP occurred 144 h post *Leishmania* infection in the AtbC-treated group, in the late phase of infection when parasites migrate to the anterior part of the digestive tract and colonize the stomodeal valve. The modulation of defensin genes also occurred in *L. longipalpis*, where both of them were reported to be modulated after the parasitic infection. LlDef1 was reduced after *L. mexicana* [65], and LlDef2 was increased after *L. infantum* infection [19]. In addition, the *P. duboscqi* defensin, which is more similar to LlDef2, was increased by *L. major* infection [18]. In *P. papatasi*, the PPAI004256 defensin, more similar to LlDef4, had different fold changes in gene expression after *L. major* or *L. donovani* infection [66]. Variability of vector’s immune responses according to different parasite species was previously described in *Anopheles* mosquitoes. The IMD-mediated response was the most effective against *Plasmodium falciparum*, while the Toll-mediated response was more effective against *Plasmodium berghei* [67,68,69,70]. It is possible that the sand fly immune response also adjusted to the different *Leishmania* species, thus adding another range of molecular events that will interfere with the sand fly permissiveness or restrictiveness in hosting other parasite species.

The parasite increase on the third day post infection reflects the natural multiplication of procyclic and its transformation into nectomonad promastigotes, which happens before the termination of blood digestion and elimination of gut contents by defecation (reviewed by [4,71]). Under our colony conditions, the *P. papatasi* blood digestion and defecation processes ended between the fourth and fifth days post blood ingestion [58]. These results indicate that the parasite infection in our experimental setting followed the commonly observed pattern, thus indicating that the parasite cycle was adjusted to changes in the sand fly microbial gut community.

To determine whether the sand fly immune response would be altered by the additive effect of *Leishmania* infection and restored gut bacteria, we removed the AtbC treatment during infection and from the sucrose meal in a second experimental group, thus allowing bacteria to multiply and recolonize the sand fly gut. There is a possibility that part of the bacterial community originates in adults from strains carried transstadially from larval and pupal stages [55]. Nevertheless, most are acquired from environmental bacteria that opportunistically colonize their hosts’ guts (reviewed by [72]). Sand flies probe surfaces with their mouthparts during their feeding process, and this habit allows new microbes to be ingested and colonize the sand flies’ guts. These studies support the hypothesis that the gut environment would be readily recolonized after antibiotics are metabolized by sand flies.

Indeed, we observed an increase in bacteria based on CFU calculation and qPCR detection in the second sand fly group where the AtbC treatment was interrupted during the experimental infection. This increase indicates that bacteria were reintroduced by sucrose feeding and surface probing or they resisted the AtbC-treatment and regrew in the sand fly gut. In this additional experimental setting, dorsal was slightly but not significantly increased, as seen in the fully AtbC-treated group. The relish expression was not significantly altered, but there was a slight increase at 72 h post feeding, indicating an earlier activation of the IMD pathway. Attacin had a highly variable and slightly increased expression at 72 h post feeding in the recovered-bacteria group. This variable expression occurred one day later than what was observed in the AtbC-treated group. This delayed attacin expression may reflect *L. major* infection reducing the pace of bacteria recolonization as similarly occurred in *L. longipalpis* infected by *L. mexicana*, which protected the sand fly from the entomopathogenic *Serratia marcescens* [73]. These findings indicate that both parasites and bacteria face a certain level of competition to survive in the sand fly gut. Possible beneficial outcomes may occur, but this complex interaction is not yet fully explored.

Interestingly, defensin increased earlier in the bacteria-recovered group, indicating that the combination of the parasite with bacteria resulted in its earlier regulation. The upregulation of defensin was observed at a time when relish was slight increased, which corresponds with the findings in AtbC-treated females about a possible connection between these two molecules. Simultaneously, the parasite detection at 72 h was variable and slightly increased, reflecting the complex and dynamic balance within the gut microbiota. For example, the reintroduction of *Lysinibacillus* or *Serratia* strains in *L. longipalpis* infected by *Leishmania chagasi* (syn. *L. infantum*) reduced the number of parasites. At the same time, *Pseudocitrobacter* had no deleterious effect on *Leishmania amazonensis* infection in the sand fly gut [74]. Our results indicate that the combination of increased bacteria and *L. major* resulted in an earlier modulation of *P. papatasi* immunity in the gut. It is possible that bacterial growth was detected as a more imminent threat, thus inducing the AMPs gene expression. These results reflect the complex dynamics between sand fly immunity and gut-residing organisms.

We plotted the relative gene expression of AtbC-treated-infected (EG1) and the bacteria-recovered (EG2) sand fly groups for comparison purposes. This alternative way of visualizing our results revealed that in the context of *L. major*-infected sand flies, the addition of bacteria did not cause a significant change in the *P. papatasi* immune response, suggesting that the *L. major* infection buffered the response against bacteria or protected the sand fly from pathogenic bacterial regrowth. This finding correlates with the protection created by *L. infantum* against *S. marcescens* infection in *L. longipalpis* mentioned above [73]. It is possible that the parasite produces molecules that compete for sand fly receptors in the gut or secrete virulent factors that would inhibit part of the bacterial community.

Some AMPs can be expressed in the fat body of insects and secreted into the hemolymph [75,76]. In addition, others can be expressed in specific tissues [70] constitutively or due to an injury-type of stimuli [77]. In *Drosophila*, the gut immune response is mediated by the IMD and Jak-STAT pathways, but AMPs can be expressed in the gut or systemically [78]. In *P. duboscqi*, a defensin peptide was identified in the hemolymph [18], while in *L. longipalpis*, defensin mRNAs were expressed in reproductive organs [79], female guts [80], or in female whole-body samples [19,65]. In addition, a *L. longipalpis* defensin gene was expressed differentially depending on the route of infection [65]. Especially for hematophagous insects, the digestive track is mostly exposed to pathogens that may be present in the ingested blood.

In the current study, among the two investigated AMPs, we identified a defensin gene exclusively expressed in gut tissue in larvae and adult females, indicating that this defensin is responsible for the specific protection of the digestive tract of *P. papatasi*. Furthermore, the defensin gene expression was upregulated by *L. major* infection. An additional load of bacteria can trigger an earlier peak of expression before completing the sand fly digestion process. The moment before defecation is strategic for parasite survival because parasites are multiplying and preparing to colonize the sand fly gut.

Very little is known about the action of defensins against *Leishmania*. One interesting study using a plant defensin PvD1 from *Phaseolus vulgaris* showed inhibitory activity against *L. amazonensis* [81]. In in vitro assays, this plant defensin inhibited promastigote proliferation and caused cytoplasmic fragmentation, the formation of multiple cytoplasmic vacuoles, and cell membrane permeabilization [81]. Although we do not discard any potential antiparasitic effects of defensins, the *P. papatasi* defensin promotor sequence is a very interesting candidate for coupling to a foreign gene coding for a parasite-killing molecule. Such a molecular construct could lead to efficient expression of an anti-leishmanial molecule in a gene-edited *P. papatasi.* Such a molecular construct may lead to efficient strategies to control parasite survival inside the vector’s gut.

## Figures and Tables

**Figure 1 microorganisms-09-02307-f001:**
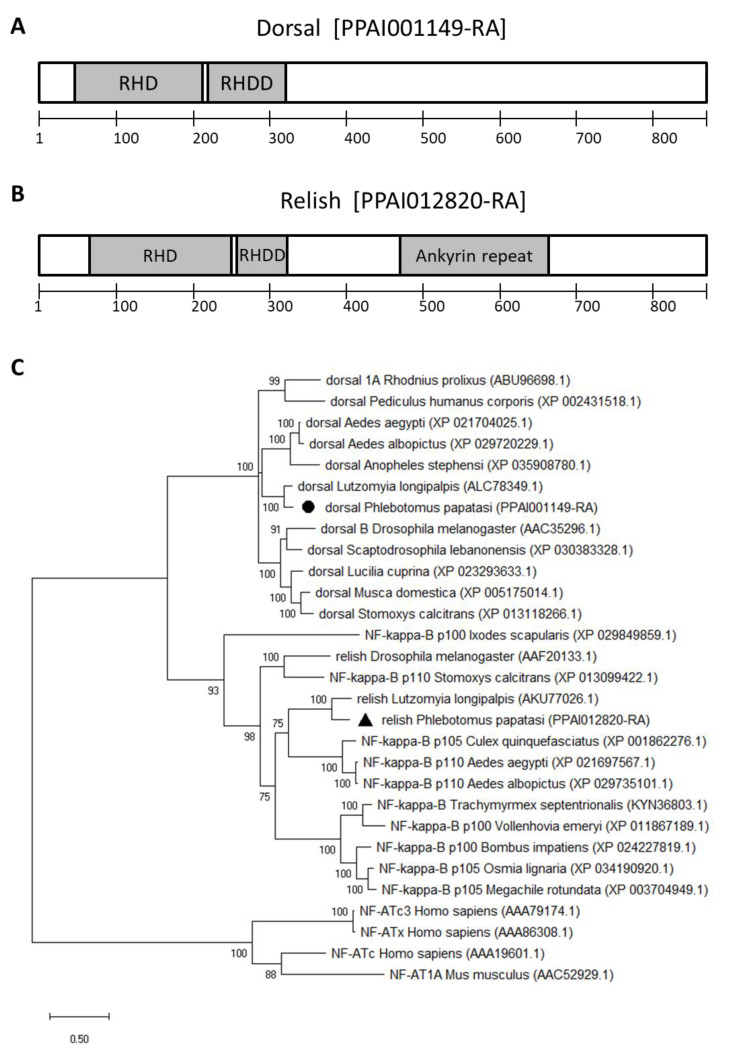
*P. papatasi* dorsal and relish amino acid sequences. (**A**,**B**)- Signature domains identified on the amino acid sequence: grey boxes indicate the rel homology domain (RHD), rel homology dimerization domain (RHDD), and ankyrin repeat domain; numeric scales indicate amino acid positions. (**C**)- Phylogram of amino acid sequences from *P. papatasi* and other organisms’ transcription factors inferred by the Maximum Likelihood method, with the WAG model and Gamma distribution. Mammalian nuclear factor of activated T-cells (NF-AT) sequences were used as an outgroup. Numbers on branch nodes indicate bootstrap values higher than 50 %. *Phlebotomus papatasi* dorsal and relish sequences are indicated by a black circle and triangle, respectively. Species names are followed by corresponding VectorBase (*P. papatasi*) or GenBank (other species) accession numbers; the scale bar indicates the number of substitutions per site.

**Figure 2 microorganisms-09-02307-f002:**
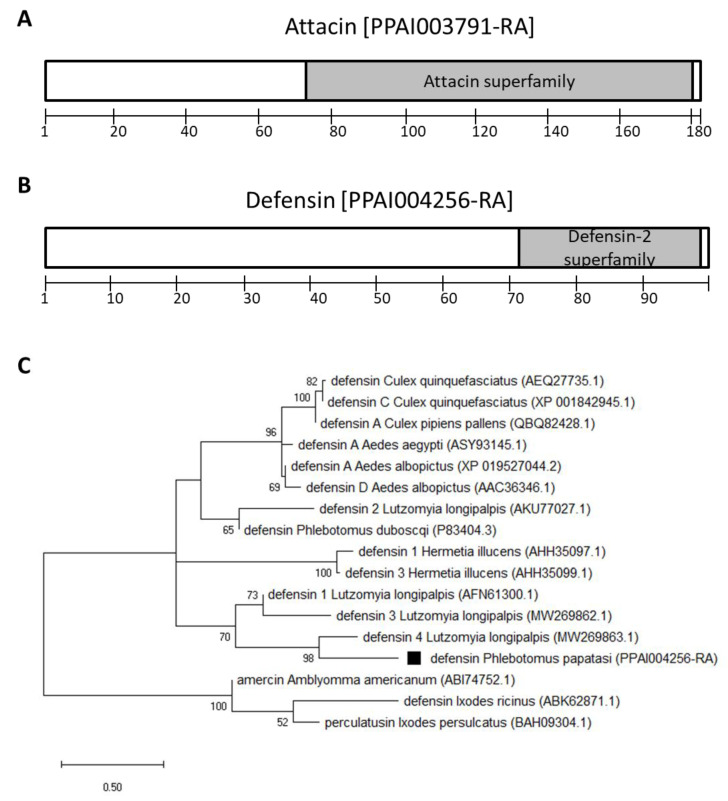
*P. papatasi* AMP amino acid sequences. (**A**,**B**)- Signature domains identified on the amino acid sequences: grey boxes indicate AMP superfamily domains; numeric scales indicate amino acid positions. (**C**)- Phylogram of defensin amino acid sequences from *P. papatasi* and other arthropods inferred by the Maximum Likelihood method, with the WAG model and Gamma distribution. Tick AMPs sequences were used as an outgroup. Numbers on branch nodes indicate bootstrap values higher than 50 %. The *Phlebotomus papatasi* defensin sequence is indicated by a black square; species names are followed by corresponding Vector Base (*P. papatasi*) or GenBank (other species) accession numbers; the scale bar indicates the number of substitutions per site.

**Figure 3 microorganisms-09-02307-f003:**
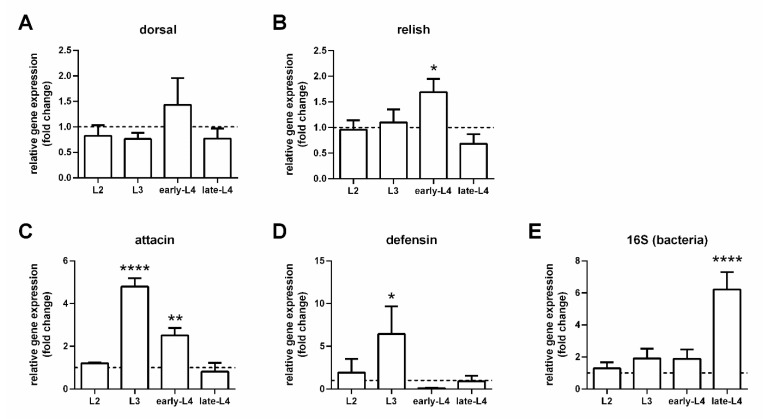
Relative gene expression of *P. papatasi* immunity genes and bacteria detection in dissected guts of larvae fed microbe-rich food. (**A**) Dorsal; (**B**) relish; (**C**) attacin; (**D**) defensin; (**E**) bacteria 16S rRNA. The y-axis represents the relative gene expression of larvae fed microbe-rich food plotted as fold change values compared to the control group fed autoclaved food (dotted line). The x-axis indicates larval stages. Vertical bars represent the average values of three independent experiments, and error bars represent the standard error. Two-way ANOVA was conducted to determine significant differences (* *p* < 0.05; ** *p* < 0.01; **** *p* < 0.0001).

**Figure 4 microorganisms-09-02307-f004:**
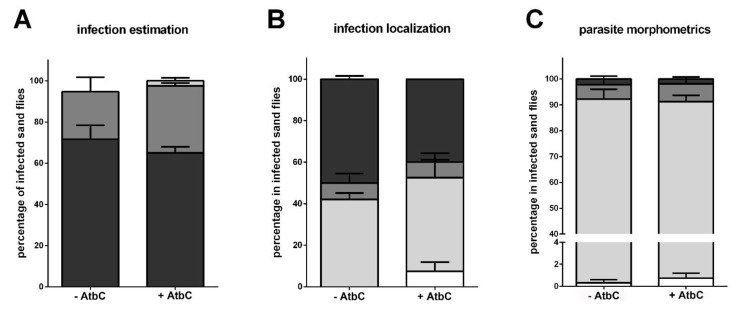
*Leishmania* infection intensity, localization, and development on the sixth day. (**A**) Infection intensity estimation. The y-axis represents the percentage of all individually inspected insects (a total of 40 sand flies in each group). Bar colors indicate infection intensity: light (light grey), moderate (mid grey), heavy (dark grey). (**B**) Infection progression in the sand fly gut. The y-axis represents the percentage of infected insects. Bar colors indicate sand fly gut localization: abdominal gut (white); thoracic gut (light grey); cardia (mid grey); stomodeal valve (dark grey). (**C**) Parasite development in the sand fly gut. The y-axis represents the percentage of analyzed parasites. Bar colors indicate parasite developmental forms: procyclic promastigote (white); elongated nectomonad (light grey); leptomonad (mid grey); metacyclic promastigote (dark grey). The x-axis represents AtbC-treated (+AtbC) and non-treated (-AtbC) groups. Vertical bars represent the average values of two independent experiments, and error bars represent the standard error. No significant differences were found (two-way ANOVA).

**Figure 5 microorganisms-09-02307-f005:**
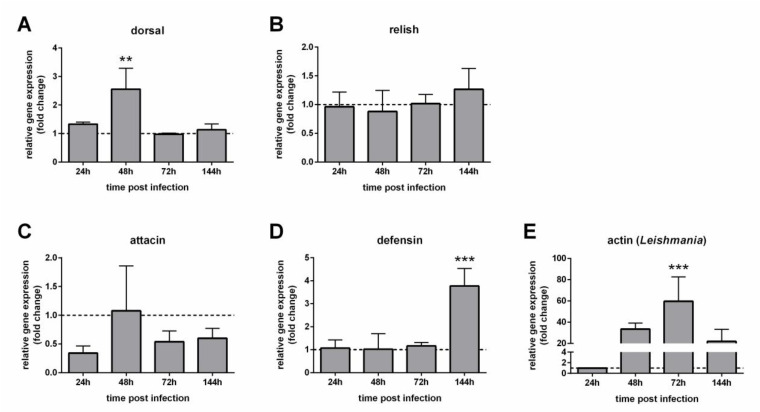
Relative gene expression of *P. papatasi* immunity genes in *Leishmania*-infected females with depleted gut bacteria (EG1) (**A**) Dorsal; (**B**) relish; (**C**) attacin; (**D**) defensin; (**E**) *Leishmania* actin. The y-axis represents relative gene expression as fold change values of *Leishmania*-infected females treated with AtbC (EG1) in comparison to the non-infected control group also treated with AtbC (dotted line) collected at the corresponding time points (**A**–**D**); *Leishmania* detection was expressed in comparison to 24 h (**E**). The x-axis indicates females collected at different times post infection. Vertical bars represent the average values of three independent experiments, and error bars represent the standard error. Two-way ANOVA was performed to determine significant differences (** *p* < 0.01; *** *p* < 0.001).

**Figure 6 microorganisms-09-02307-f006:**
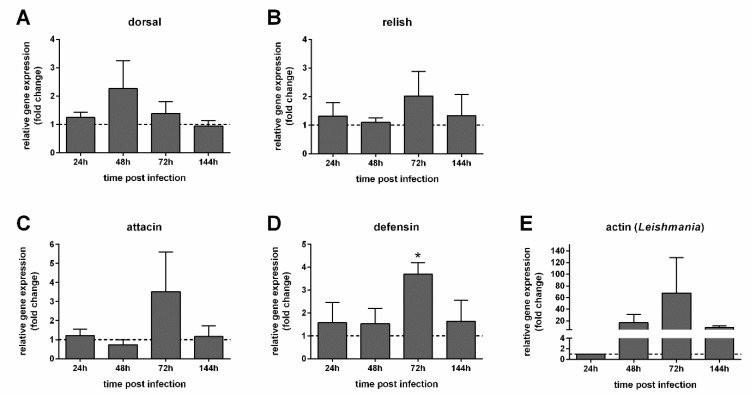
Relative gene expression of immunity genes in *Leishmania*-infected females with recovered gut bacteria (EG2). (**A**) Dorsal; (**B**) relish; (**C**) attacin; (**D**) defensin; (**E**) *Leishmania* actin. The y-axis represents the relative gene expression as the fold change values of *Leishmania*-infected females with recovered gut microbiota (EG2) in comparison to the non-infected control group also treated with AtbC (dotted line) collected at the corresponding time points (**A**–**D**); *Leishmania* detection was expressed in comparison to 24 h (**E**). The x-axis represents females collected at different time points post infection. Vertical bars represent the average values of three independent experiments, and error bars represent the standard error. A two-way ANOVA was performed to determine significant differences (* *p* < 0.05).

**Figure 7 microorganisms-09-02307-f007:**
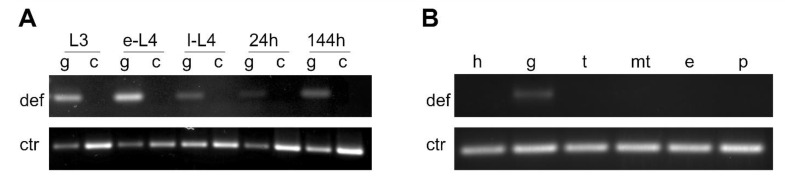
PCR amplification of *P. papatasi* defensin gene in different tissues. (**A**) Defensin PCR from guts (g) and carcasses (c) from L3, early L4 (e-L4), late L4 (l-L4) larval stages, and from females collected 24 h and 144 h post infection. (**B**) Defensin PCR from heads (h), guts (g), thorax (t), Malpighian tubules (mt), eggs (e), and posterior end of the abdomen of females collected on day four (96 h) post blood feeding. Representative images of electrophoresis of defensin (def) and control (ctr) PCR products in 1.5 % agarose gel are shown.

**Table 1 microorganisms-09-02307-t001:** AtbC treatment and *Leishmania* experimental infection.

Sand Fly Groups	AtbC in Sucrose Meal before Blood Meal	AtbC in Blood Meal	*Leishmania* in Blood Meal	AtbC in Sucrose Meal after Blood Meal
Control group	+	+	-	+
Experimental group 1 (EG1)	+	+	+	+
Experimental group 2 (EG2)	+	−	+	−

## Data Availability

Nucleotide and amino acid sequences used in this study can be found in NCBI GenBank (https://www.ncbi.nlm.nih.gov/ accessed on 5 November 2021) and VectorBase (https://vectorbase.org/vectorbase/app/ accessed on 5 November 2021). Original gel images of Figure 7 were uploaded in FigShare (https://doi.org/10.6084/m9.figshare.16939591.v1).

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
