# Peer review of "Phlebotomus papatasi Antimicrobial Peptides in Larvae and Females and a Gut-Specific Defensin Upregulated by Leishmania major Infection"

_microorganisms, 2021, doi:10.3390/microorganisms9112307_

Round 1
Reviewer 1 Report
This study was aimed to investigate how Phlebotomus papatasi immunity-related genes were regulated against Leishmania major and commensal bacteria. The authors focused on the transcriptional factors, dorsal and relish, and antimicrobial peptides, attacin and defensin, and analyzed their expression changes by qPCR. The authors concluded that a gut-specific defensing upregulated by L. major infection, in combination with gut bacteria, is a promising molecular target for parasite control. The result is interesting as a basic study for sand fly immunity against Leishmania infection. Please consider following comments.
- Materials and methods, 2.9.: I strongly recommend to specify the P. papatasi blood digestion and defecation processes end between the fourth- and fifth-days post blood ingestion in this part, as mentioned in Discussion (lines 442-443), since it is very important information why the authors used sand flies of day 6 post infection.
- Didn’t you find any other homologues of dorsal, relish, attacin, and especially defensin genes identified in the sand fly used in this study? Since Lutzomyia longipalpis is shown to have 4 defensins, and P. duboscqi defensin located another clade from that of P. papatasi in spite of genetically very closely-related, I wonder if P. papatasi has other defensins. In addition, did you confirm the sequences of these genes in your own sand fly since the gene sequences may somehow different among strains?
- The authors compared relative gene expression of P. papatasi immunity-related genes between groups in the presence and absence of bacteria in their food. Are there any other factors to be considered affecting the results, such as nutrient? How was the difference of growth period, survival and size of larvae and adults between groups? These information is important because nutrient status can affect the host immunity.
- I did not understand very well about the objective of figure 5. As mentioned by the authors, sand flies at 24h, 48h, and 72h post blood meals still has blood in peritrophic matrix (PM) in the midgut. Therefore, parasites grow in PM at these points, like a growth in the culture medium (Fig. 5E). In addition, very few parasites could access to the midgut at these points, and I think expression changes of immunity-related genes cannot be expected. The data of 144h post infection is meaningful.
- Discussion should be reorganized. Mostly, it is just repeats of the present results followed by previous reports. The results should be well-discussed in response to previous reports in Discussion.
- The main result is a gut-specific defensin upregulated by L. major infection is a promising molecular target for parasite control. Defensins are well-known to disrupt microbial membranes since membrane structure of prokaryote is quite different from that of eukaryote. I wonder how defensins effect on the control of Leishmania. It should be discussed in Discussion.
- Lines 472-474: I agree partly, but another possibility should be considered. After removing antibiotics, bacteria must start growing. I think antimicrobial peptides were induced in response to growing bacteria rather than Leishmania.
Author Response
We thank reviewer#1 for the constructive criticism.
Please find below our point-by-point response in bold letters.
Materials and methods, 2.9.: I strongly recommend to specify the P. papatasi blood digestion and defecation processes end between the fourth- and fifth-days post blood ingestion in this part, as mentioned in Discussion (lines 442-443), since it is very important information why the authors used sand flies of day 6 post infection.
We followed reviewer#1 recommendation and included this information in the beginning of section 2.9. (lines 158 to 161).
Now it reads as: “Leishmania development in sand fly vectors was examined by light microscopy in 20 sand fly guts 144 h post-infection (day 6 PI), a time after blood meal was digested and defecated. In conditions used, most P. papatasi females defecate on day 4 PI [30].”
Didn’t you find any other homologues of dorsal, relish, attacin, and especially defensin genes identified in the sand fly used in this study? Since Lutzomyia longipalpis is shown to have 4 defensins, and P. duboscqi defensin located another clade from that of P. papatasi in spite of genetically very closely-related, I wonder if P. papatasi has other defensins. In addition, did you confirm the sequences of these genes in your own sand fly since the gene sequences may somehow different among strains?
When using L. longipalpis genes as query for blast search against the P. papatasi database, we found one homologue for attacin (PPAI003791) and relish (PPAI012820). For dorsal, we retrieved two different genes: PPAI001149 (coding for a protein with 865 amino acids) and PPAI001148 (coding for a protein with 206 amino acids). Considering that most of the dorsal proteins described in Aedes, Anopheles, and Drosophila species have around 600 amino acids or more, we decided to focus our efforts on the PPAI001149 gene at this time. For defensins, we found two different genes that showed top best hits with L. longipalpis defensins: PPAI004256 (similar to LlDef1, LlDef3, and LlDef4) and PPAI010650 (similar to LlDef2). We tried a considerable large set of primer pairs aiming the PPAI010650 coding sequence, but we failed to design a primer pair that could specifically target PPAI010650 and have an efficiency score above 90%. Therefore, we chose to work with PPAI004256 at this time.
We designed gene-specific primers within each of the gene coding sequences for PCR amplifications. Amplicons were sequenced for confirming the target gene. In our current approach, we were not able to distinguish differences between P. papatasi sequences derived from VectorBase (originated from Israel) and from our colony (originated from Turkey). Nevertheless, reviewer#1 question is very interesting and worth pursuing in the future. This information was included in the results section (lines 225 to 229).
The authors compared relative gene expression of P. papatasi immunity-related genes between groups in the presence and absence of bacteria in their food. Are there any other factors to be considered affecting the results, such as nutrient? How was the difference of growth period, survival and size of larvae and adults between groups? These information is important because nutrient status can affect the host immunity.
What supported us on making comparisons between larvae groups fed on autoclaved or microbe-rich food was the fact that both food types were originated from the same batch, thus having the same initial nutrient composition. But as digestion occur in the larvae gut, we cannot exclude possible effects on nutrient processing derived from differences in microbial composition between the two larval groups.
Unfortunately, we did not assess the nutrient composition of the food prior offering it to the larvae in our experiments. But we agree with reviewer#1 that nutrient availability could affect the sand fly larvae immunity. We did not measure individual larvae or emerged adults sizes, but we have records of larvae development done during larvae maintenance. We included a short description of corresponding methods (lines 101 to 102), results (lines 223 to 237), and a short statement in our discussion (lines 393 to 398) addressing the aspect raised by reviewer#1. A supplementary table (Table S1) was included with the elapsed time between larval stages L3 developing into early-L4, and the time between early-L4 to late-L4 obtained from two independent experiments.
Sentences included in methods section reads as: “Observation of larvae development and emerged adults was recorded following colony maintenance routine also three times a week.”
Sentences included in results section reads as: “The larvae growth period was slightly delayed in the group fed on autoclaved food be-tween L3 and early-L4 stages, but no difference was observed between early-L4 and late-L4 stages (Table S1). Nevertheless, there was no noticeable differences in larvae size and survival rates, neither observed in size or number of emerged adults between the two groups of larvae rearing.”
Sentences included in discussion section reads as: “We cannot exclude possible effects on larvae immunity derived from nutrient processing or absorption caused by the differences in microbial composition in the food [50]. Nevertheless, our experimental approach is supported by the fact that both food types were originated from the same batch, thus having the same initial nutrient composition. In addition, there was no noticeable difference in development between the two larvae groups.”
I did not understand very well about the objective of figure 5. As mentioned by the authors, sand flies at 24h, 48h, and 72h post blood meals still has blood in peritrophic matrix (PM) in the midgut. Therefore, parasites grow in PM at these points, like a growth in the culture medium (Fig. 5E). In addition, very few parasites could access to the midgut at these points, and I think expression changes of immunity-related genes cannot be expected. The data of 144h post infection is meaningful.
Indeed, between 24h and 72h post blood meal, it is expected that parasites are enclosed in the PM. However, we cannot rule out that parasite secreted molecules and exosomes (Atayde et al., 2015) (https://pubmed.ncbi.nlm.nih.gov/26565909/ ) could pass through the PM, especially while not fully formed or starting to be degraded. Because of these possibilities we did not refrain from investigating the expression the sand fly genes in 24h, 48h and 72h post blood ingestion.
We added a paragraph in our discussion section to clarify this topic (lines 448 to 455).
Discussion should be reorganized. Mostly, it is just repeats of the present results followed by previous reports. The results should be well-discussed in response to previous reports in Discussion.
We improved our discussion section (lines 367 to 368, 429 to 431, 445 to 447, 460 to 463, 469 to 474, 482 to 485, 490 to 493, 502 to 504, 517 to 520, 522 to 525, 542 to 543, and 551 to 553).
The main result is a gut-specific defensin upregulated by L. major infection is a promising molecular target for parasite control. Defensins are well-known to disrupt microbial membranes since membrane structure of prokaryote is quite different from that of eukaryote. I wonder how defensins effect on the control of Leishmania. It should be discussed in Discussion.
Indeed, defensins activities against prokaryotic cell such as creating channels, disruption of osmotic balance and metabolism are well characterized. On the other hand, very little is known about their action against Leishmania. One interesting study done by Nascimento et al. (2015) (https://pubmed.ncbi.nlm.nih.gov/26285803/ ) using a plant defensin PvD1 from Phaseolus vulgaris showed an inhibitory activity against L. amazonensis. In in vitro assays, this plant defensin inhibited promastigotes proliferation, caused cytoplasmic fragmentation, formation of multiple cytoplasmic vacuoles, and cell membrane permeabilization.
Regarding sand fly defensins, the study done by Boulanger et al. (2004) (https://pubmed.ncbi.nlm.nih.gov/15557638/ ) in P. duboscqi showed an antiparasitic activity of a recombinant protein, but its mode of action is still not known.
Although we do not discard any antiparasitic potentials of defensins, our conclusion on the importance of a gut specific defensin upregulated by L. major infection doesn’t assume its potential killing effect on parasites. We consider that the main relevance relies on the potential use of its regulatory region. Its promotor sequence could be coupled to a foreign gene coding for a parasite killing molecule. Such molecular construct may lead to efficient expression of an anti-leishmanial molecule in a gene-edited P. papatasi during critical phases of the parasite development. This is a very interesting field, and we are currently aiming our future investigations in this direction.
We added a new closing paragraph reflecting this topic (lines 561 to 572).
Lines 472-474: I agree partly, but another possibility should be considered. After removing antibiotics, bacteria must start growing. I think antimicrobial peptides were induced in response to growing bacteria rather than Leishmania.
Our data show that by removing antibiotics both parasite and bacteria become abundant. It is difficult to precisely argue which of them is the main cause of the sand fly immune response after both become abundant in the sand fly gut. On the other hand, there is a possibility that the growing of bacteria may be detected as a more imminent threat.
We rephrased our text to address the point raised by reviewer#1 (lines 530 to 534). Now it reads as: “Our results indicate that the combination of increased bacteria and L. major resulted in an earlier modulation of P. papatasi immunity in the gut. It is possible that the growing of bacteria was detected as a more imminent threat, thus inducing the AMPs gene expression. These results reflect the complex dynamics between sand fly immunity and gut residing organisms.”
Reviewer 2 Report
x
Author Response
We did not have access to comments from reviewer#2.
Reviewer 3 Report
After reading the manuscript “Phlebotomus papatasi antimicrobial peptides in larvae and females, and a gut-specific defensin upregulated by Leishmania major infection”, I have found it is an interesting manuscript.
However, in my opinion the authors should
-In Material and methods section
--Please add composition of bacterias in “microbe-rich food”
-- Is the composition of “autoclaved food” the same that “microbe-rich food”???
-- In Table 2: Oligonucleotides, please write the nucleotide sequence correctly (5´……..3´)
In Results section
-In Figure 3 authors show “Relative gene expression of P. papatasi immunity genes and bacteria detection in dissected guts of larvae fed on microbe-rich food”. Are gene expression the same in all larval stages when they feed autoclaved food? this question is not clear in the text
- Have the authors analyzed the number of metacyclic promastigotes?
Author Response
We thank reviewer#3 for the positive criticism.
Please find our point-by-point response, highlighted in bold letters:
-In Material and methods section
--Please add composition of bacterias in “microbe-rich food”
The larvae food used in our experiments was prepared from aerobically composted rabbit feces. This process was described in detail by Lawyer et al. (2017) (https://pubmed.ncbi.nlm.nih.gov/29139377/). The microbe-rich food is the composted rabbit food kept unaltered (not sterilized). We did not seed any additional bacteria strain to it. Nevertheless, the composition of bacteria in this mixture is not available because we did not make bacteriome identification.
For clarity, we added sentences to our methods referring to the reviewer#3 comment (lines 94 to 99). Now it reads as: “Larval food made from composted rabbit feces [26,27] was divided into two parts, one part was sterilized in an autoclave, and another part was kept unaltered, hence forth referred as microbe-rich food. Both were collected from the same batch of composted food; therefore, they have same initial composition. We did not make the bacteriome identification in the types of larvae food used in our experiments.”.
-- Is the composition of “autoclaved food” the same that “microbe-rich food”???
Both autoclaved and microbe-rich food were collected from the same batch of composted food. The difference between them is that one was sterilized while the other was left unaltered. We clarified the comments on food composition as described in our answer to reviewer request above.
-- In Table 2: Oligonucleotides, please write the nucleotide sequence correctly (5´……..3´)
Corrected
In Results section
-In Figure 3 authors show “Relative gene expression of P. papatasi immunity genes and bacteria detection in dissected guts of larvae fed on microbe-rich food”. Are gene expression the same in all larval stages when they feed autoclaved food? this question is not clear in the text
Samples of larval stages collected from the group fed on autoclaved food were used as control samples for their corresponding larval stages fed on the microbe-rich food. Therefore, for each stage, the gene expression of larvae fed on autoclaved food are represented as fold change 1 (dotted line). Nevertheless, within larval stages collected from the same feeding regimen it is expected to have some differential gene expression as reported in L. longipalpis larvae immune response by Telleria et al. (2021).
We corrected our qPCR methods (lines 99, 155-156).
- Have the authors analyzed the number of metacyclic promastigotes?
Yes, the numbers were estimated based on methodology described in chapter 2.9. “Leishmania infection estimation and morphometrics”. The results are shown in Figure 4 - C “Parasite development in the sand fly gut” where the numbers are presented as a percentage of developmental forms from the total of analyzed parasites (metacyclic promastigotes are shown in dark grey color).
Round 2
Reviewer 3 Report
In the new versión, the authors have satisfactorily responded to all my questions and made the necessary changes to the manuscript so the article has improved substantially.
Author Response
We thank reviewer#3 for the positive criticism that helped to improve our manuscript.